# ESM-Tools Version 4.0: A modular infrastructure for stand-alone and coupled Earth System Modelling (ESM)

Dirk Barbi[1], Nadine Wieters[1], Paul Gierz[1], Fatemeh Chegini[1, 3], Sara Khosravi[2], and Luisa Cristini[1]

[1]Alfred Wegener Institute Helmholtz Center for Polar and Marine Research, Bremerhaven, Germany
[2]Alfred Wegener Institute Helmholtz Center for Polar and Marine Research, Potsdam, Germany
[3]Max Planck Institute for Meteorology, Hamburg, Germany

**Correspondence:** Luisa Cristini (luisa.cristini@awi.de)

**Abstract.**

Earth system and climate modelling involves the simulation of processes on a wide range of scales and within and across various compartments of the Earth system. In practice, component models are often developed independently by different research groups, adapted by others to their special interests, and then combined using a dedicated coupling software. This procedure not only leads to a strongly growing number of available versions of model components and coupled setups, but also to model- and HPC-system dependent ways of obtaining, configuring, building, and operating them. Therefore, implementing these Earth System Models (ESMs) can be challenging and extremely time consuming, especially for less experienced modellers, or scientists aiming to use different ESMs as in the case of inter-comparison projects.

To assist researchers and modellers by reducing avoidable complexity, we developed the ESM-Tools software, which provides a standard way for downloading, configuring, compiling, running and monitoring different models on a variety of High Performance Computing (HPC) systems. It should be noted that the ESM-Tools are equally applicable and helpful for stand-alone as for coupled models. In fact, the ESM-Tools are used as standard compile and runtime infrastructure for FESOM2, and currently also applied for ECHAM and ICON standalone simulations. As coupled ESMs are technically the more challenging tasks, we will focus on coupled setups, always implying that stand-alone models can benefit in the same way.

With the ESM-Tools, the user is only required to provide a short script consisting of only the experiment specific definitions, while the software executes all the phases of a simulation in the correct order. The software, which is well documented and easy to install and use, currently supports four ocean models, three atmosphere models, two biogeochemistry models, an ice sheet model, an isostatic adjustment model, a hydrology model and a land-surface model. Compared to previous versions, ESM-Tools has been entirely re-coded in a high level programming language (Python) and provides researchers with an even more user-friendly interface for Earth system modelling lately. The ESM-Tools were developed within the framework of the project Advanced Earth System Model Capacity, supported by the Helmholtz Association.

## 1 Introduction

Earth System Models (ESMs) are widely used for studying past, present and future climates, processes in the different Earth system compartments (e.g., atmosphere and ocean), as well as the interactions between them. Therefore, ESMs can include

several model components (e.g. atmosphere, ocean, ocean biogeochemistry, land and sea ice, land biosphere, hydrology) which are developed independently by different research groups. Various ESMs have been developed in recent years by using different model components and couplers. Therefore, in order to be able to apply these models, the users need to acquire knowledge of various model and system dependent parameters. This often proves time consuming and challenging for scientists who are new to numerical modelling or are approaching a numerical model they are not familiar with. Since the ESMs are usually written for a specific purpose that e.g. require a specific coupling and/or tailored ESM components, they often lack modularity. This lack of modularity in coupled systems can make it very inefficient when it comes to replacing certain model components. In the context of further HPC development, e.g. the full use of emerging heterogeneous HPC systems, the use of coupled model systems becomes particularly complex (see also Bauer et al. (2015), Schulthess et al. (2019)). This issue can be further enhanced by lack of training in software development, often (but not always) amongst early career scientists, making the procedure non-trivial and error prone. To address these issues, we developed the ESM-Tools — a software that considerably reduces the difficulty of applying ESMs, by providing a modular external modelling infrastructure that allows scientists to work with standalone as well as coupled setups in a very intuitive and straightforward manner.

The complete workflow for ESM applications includes tasks such as obtaining and compiling all source codes, managing input data and configuration files for model setup, submitting the executable to a multi-processor system, monitoring and logging the process and managing/post-processing output data. These tasks require knowledge of the parameters used by the ESM such as build environment, configuration files (e.g. Fortran namelists) and I/O structure as well as knowledge of the specific High Performance Computing (HPC) environment (e.g. installed libraries and batch system). This often results in an ESM modeller having to deal with numerous technical issues. The ESM-Tools software provides standard solutions to typical tasks occurring within the workflow of Earth system modelling, such as calendar operations, data post-processing and monitoring, sanity checks, sorting and archiving of output, and script-based coupling. The ESM-Tools facilitate Earth system modelling by providing a standardized framework to download, configure, compile, run, and analyze/monitor a variety of ESMs on various HPC systems.

A variety of software and tools have been developed to assist Earth system modellers with running coupled models. However, most of them have been developed for specific purposes and lack adaptability.

– Probably closest to the ESM-Tools, both by functionality and code design, is the *ScriptEngine*[1] simulation task generating software developed at SMHI primarily for the EC-Earth community. Indeed, ScriptEngine and ESM-Tools both use a combination of YAML configuration files and Python methods, but while the ESM-Tools use YAML to store the information about models, setups etc., ScriptEngine goes more into the direction of defining a Domain Specific Language (DSL, see also Lawrence et al. (2018)), allowing the user to describe commands and simple routines in YAML format. This approach also is very elegant and leads to a simple and user-friendly way to formulate what used to be run scripts. To our knowledge, it is not in an advanced state yet, and only contains functions for the EC-Earth model at this point. Also to our understanding it does not contain download and compile functionality, as well as a natural concept of modularity

---

[1]https://pypi.org/project/scriptengine

of model components. We hope to be able to cooperate with the ScriptEngine developers in the future, as both software package can be seen as natural extensions of each other.

- The *Modular Earth Submodel System*, MESSy[2], has been developed as an infrastructure with generalized interfaces for implementation of ESMs (Jöckel et al., 2005). The focus of MESSy, even though it comes with its own scripts and tools for compiling and running, is internal infrastructure, meaning the definition of reusable codes ("building blocks", "dwarfs", ...) as operators, automatic management of memory layout, and internal 3-D coupling of fields through all compartents of a coupled setup. MESSy and ESM-Tools don't interfere at all, as one is for the internal, and the other for external infrastructure, and can in fact be used together. We already profit from a frequent exchange and cooperation with the MESSy developers in the context of the Helmholtz-project ESM, and have implemented first steps towards integrating full MESSy support into the ESM-Tools, hoping to complete this process by release 5.0 (planned for October 2020).

- The *Modular System for Shelves and Coasts*, MOSSCO[3] provides a modular system for domain and process coupling of applications in the coastal ocean (Lemmen et al., 2017). It is designed to enable integrated regional coastal modelling and is targeted towards the coupling of model components with different orders of magnitude of spatial and temporal resolution. The framework allows for the seamless replacement of individual model components. In contrast to the ESM-Tools, MOSSCO is a coupling framework rather than a framework for building and running ESMs.

- The *Make Experiments* tools (*Mkexp*[4]) contain a set of tools for preparing experiments with the Earth system models developed at the Max-Planck Institute for Meteorology. It was primarily dedicated to generating run- and postprocessing scripts for the MPI-ESM coupled Earth System Model from configuration files. One main difference between *Mkexp* and ESM-Tools is that the direct output of *Mkexp* are lengthy shell scripts, which are then submitted by the user to HPC systems. Even though *Mkexp* was successfully adapted to other coupled ESMs, lacks in our opinion the needed modularity required for optimal coupling of ESMs. The main problem of *Mkexp* seems to be that modellers don't use it as intended though, distributing and editing the very long and unintuitive shell scripts, rather than generating new ones. Out of this experience we decided that the ESM-Tools should not generate (complex) scripts, but rather interpret short and simple ones.

- The *Earth System Modeling Framework*, ESMF[5] is a standard software platform for Earth system models (Hill et al., 2004). It provides different structures for the interconnecting between model components and provides a standard support library for the construction of components. The emphasis of ESMF is to ensure a standard infrastructure of component coupling and may require code adaptation in order to fit into its framework.

---

[2]https://www.messy-interface.org/

[3]http://www.mossco.de

[4]https://code.mpimet.mpg.de/projects

[5]http://earthsystemmodeling.org/

– ESM-Tools should not be confused with the *Earth System Model Evaluation Tool* (ESMValTool[6]) which is a community diagnostics and performance metrics tool, used to compare one or several models against observations or their previous versions (Righi et al.; Eyring et al., 2015). ESM-Tools and ESMValTool are not related.

In contrast to other software described above, e.g. MESSy or ESMF, ESM-Tools are designed to help scientist to build and run different standalone models as well as coupled setups without any need of code adaptation. To ensure this, all ESM components need to have a dedicated coupling already implemented. To our knowledge, there currently is no modular infrastructure that assists modellers in operating ESMs, incorporates a variety of stand-alone and coupled systems, is extendable and flexible, and

95 is open to a larger community of researchers. The ESM-Tools fill this gap by fulfilling the criteria such as user-friendliness, modularity, portability, maintainability, extendablilty (see also Section 2.2.2).

This paper aims to present the ESM-Tools as released in version 4.0 (April 2020); it explains the objectives of the ESM-Tools, describes the development steps of the software; and provides a short overview of the purpose and usage of each of the individual tools. The paper is structured as follows: In Section 2, the ESM-Tools are described including the programming

language, their structure and supported numerical models. Section 3 explains the development of the tools with focus on software and code management. The tools application is discussed in Section 4. The summary including the benefits of using the tools and the experiences gained during the development is detailed in Section 5. Supporting information and documentation is given in Section A.

## 2 ESM-Tools Description

The technical aspects of applying an Earth System Models (ESMs) can be challenging and time consuming. This is especially true for less-experienced modellers, but also holds for highly-experienced scientists. ESM-Tools is a software developed to reduce avoidable complexity by providing an infrastructure for obtaining and operating both standalone models and coupled systems.

## 2.1 Overview

The ESM-Tools software is devided into three major parts. The first one, called *esm_tools*, is the starting point when installing the tools, and consists only of a collection of YAML configuration files containing all information on models, coupled setups, HPC systems etc., as well as the (extensive) documentation. After cloning *esm_tools*, an installer for the other packages can be used to *pip*-install the python packages containing the actual code. The whole installation process of the ESM-Tools thus consists of what can be seen in fig. (1).

The second major part, and first important executable installed, is the *esm_master*-tool that downloads, configures and compiles model components, coupled setups, libraries etc.. The tool itself is written without any reference to specific models

---

[6]https://www.esmvaltool.org/

```
git clone https://github.com/esm-tools/esm_tools.git
cd esm_tools
./install.sh
```

**Figure 1.** Installation process of the ESM-Tools. Please make sure come basic requirements are met before trying to install (like setting the locale, and providing a recent version of git).

or HPC environments, but takes the required information from the *esm_tools* configuration files, i.e. in the case of the standalone *FESOM-2*:

- Hardware and software stack settings from a machine-dependant YAML file, e.g. in the case of *mistral.dkrz.de* this file
is located at *esm_tools/configs/machines/mistral.yaml*

- Information on how *git* works from *esm_tools/configs/other_software/git.yaml*

- Repository name, default branch, compile settings, and necessary model-dependant environment changes for *FESOM-2* from a component yaml file, in this case *esm_tools/configs/components/fesom/fesom-2.0.yaml*

The information is collected and evaluated, and then used to either download, configure, or compile the model. To install
*FESOM-2*, it is thus sufficient to type *esm_master install-fesom-2.0*, provided one has the right to do so (i.e., access to the repository used for cloning). In case of a coupled setup, *esm_master* first reads the YAML configuration for the coupled setup, which specifies the components are used, and then the configurations for the (standalone) model components. In case of a conflict, the information from the coupled setup configuration overwrites the standalone configuration.

The third and most complex major part is the *esm_runscripts*-package, that takes care of the whole workflow for model
simulations. To use *esm_runscripts*, the user needs to provide a very short run configuration - the easiest way to provide this would be another YAML file, but we also still support shell scripts at this point, as most users are more acquainted to these. For *FESOM-2* for example, such a run configuration might look like fig. (2). Another runscript, in shell format, can be found in the appendix. To each supported model component and coupled setup, a number of sample runscripts are distributed with *esm_tools*.

Similar to *esm_master*, *esm_runscripts* takes the necessary information to perform the experiment from the same YAML configuration files in *esm_tools*. To start a run, the user types *esm_runscripts myrunscript.yaml -e EXPID*, where *EXPID* is the unique name of the experiment. *esm_runscripts* will then set up the experiment folder, copy / link the needed files, apply changes to namelists etc. and finally execute the simulation. There is a lot more to *esm_runscripts*, some features will be described in this paper, but to cover all of them is beyond the scope of this introductory paper. The interested reader is kindly
invited to have a look at the documentation on https://esm-tools.readthedocs.io/en/latest/ for more details.

Since its first release, the application of ESM-Tools has proven beneficial for the scientific modelling community. New team members have successfully used ESM-Tools to understand and run a standard ESM experiment in 1–2 days rather than 3–4

```
general:
        setup_name: fesom
        compute_time: "00:08:00"
        initial_date: '2001-01-01'
        final_date: '2001-03-31'
        base_dir: "/path/to/experiment/data/"
        nyear: 0
        nmonth: 1
        use_venv: False

fesom:
        version: 2.0
        model_dir: "/path/to/modelcodes/fesom-2.0/"
        pool_dir: "/path/to/input/data/"
        mesh_dir: "/path/to/FESOM/meshes/mesh_CORE2_final/"
        res: CORE2
        lresume: 0
        restart_rate: 1
        restart_first: 1
        restart_unit: 'm'
        post_processing: 0
```

**Figure 2.** Run configuration for a short run of *FESOM-2*.

weeks prior to the availability of ESM-Tools. More experienced users have run new experiments within hours. Another benefit
has been the reduction of time spent by modellers to switch to a different model or coupled system. Since ESM-Tools provides
a common infrastructure for a number of different models, the workflow does not change when changing to another model or
HPC system. This has proven to save modellers a considerable amount of time in the order of several days due to reduction of
technical work. In this sense, ESM-Tools has also the potential of reducing model diversity by making it easier for modellers
to switch to other modelling systems, without having to deal with periods of reduced scientific productivity.

In addition to increasing the time efficiency, ESM-Tools has assisted modellers in handling and managing model data.
*esm_runscripts* manages the input and output transfer of data and log files to associated folders. This is beneficial for individual
modellers as well as modelling teams. Through ESM-Tools, the modellers are able to easily exchange the path to model
input data on a specific HPC. Furthermore, applying *esm_runscripts* results in a common structure for the input/output and
log files/folders across model simulations. This expedites the exchange of data as well as solving a model run error. Model
developers and system administrators, who have added or changed certain aspects in a model component (e.g. fixed a bug,
changed configurations/parameterisations or switched pre-installed libraries with newer versions) have also used the ESM-
Tools to share their changes with all users in an efficient and standardized way.

## 2.2 Implementation and Configuration

One objective of the development of the ESM-Tools is to provide users with easy to understand functions and configuration
files. To achieve this, the software architecture of the ESM-Tools is structured so that all information that are mandatory for an

experiment (e.g., HPC systems, input datasets) are contained in separate YAML[7] configuration files, while the actual program, i.e. the commands to be performed using this information, is in itself entirely independent of the models or HPC systems used, while the ESM-Tools functionality has been fully coded in Python. This separation of information from implementation facilitates the development and maintenance of the functions while making the tools more user-friendly. Furthermore, adaption

of the tools to new model configuration files and extension of the tools to include new ESMs is made easier.

Splitting information contained in the YAML configuration files from the Python implementation of the functionality in the described way has proven to be a very robust strategy to provide a modular and extendable software tool, with limited need for maintenance as users primarily edit the configuration files or user-friendly information part (written in YAML), instead of writing additional (Python) code.

### 2.2.1 Python Implementation

As a high-level, object-oriented language, Python enables algorithm flexibility and ease of programming (Pelupessy et al., 2017). Furthermore, Python is widely used by the scientific community and has many available libraries. Therefore, it is easier for the ESM-Tools users to understand the functions and contribute to the development.

We organized our (Python) code in several independent Python packages, each can be installed either using the installer

distributed with the YAML configurations, or the well-known Python installer *pip*.

As of version 4.0 of the ESM-Tools, the Python code consists of the packages listed in Table 1. In addition to the listed Python

| Python package | Description |
| --- | --- |
| *esm_master* | Python functions of the executable *esm_master*, the tool to download, config and compile |
| *esm_runscripts* | functions for the executable *esm_runscripts*, interprets runscripts, prepares and performs simulation, sorts the output etc. |
| *esm_version_checker* | provides the executable *esm_versions*, which helps manage / update the versions of these Python packages |
| *esm_environment* | assembles the runtime or compiletime environment needed |
| *esm_parser* | parser for the YAML configuration, also performs list expansion, basic math, conditional parsing etc. |
| *esm_calendar* | basic calendar functions, much like datetime, but works for paleo time scales |
| *esm_rcfile* | writes and reads the file *~/.esmtoolsrc* |

**Table 1.** ESM-Tools Python packages

packages, we make use of a self-written plugin manager that can also be used to extend the functionality without having to

---
[7]https://yaml.org/

edit the ESM-Tools core. In detail, users and developers can write own python code, and use a YAML configuration file alter the flow of execution of *esm_runscripts*, while sticking to the very simple interface that a user-defined function should take a python dictionary as input, and also return a dictionary (basically the internal representation of all YAML files; this dict contains all information available at the moment of execution of the function). Fig. (3) shows the order of execution a compute job for example, as defined in *esm_runscripts.yaml*, where additional calls could be inserted. Changes to ESM-Tools python code are not necessary, which not only makes the development of new functionality much easier, but also helps keeping the core code more stable.

```
compute:
    recipe:
        - "venv_bootstrap"
        - "_create_setup_folders"
        - "_create_component_folders"
        - "initialize_experiment_logfile"
        - "copy_tools_to_thisrun"
        - "compile_model"
        - "_copy_preliminary_files_from_experiment_to_thisrun"
        - "_show_simulation_info"
        - "create_new_files"
        - "prepare_coupler_files"
        - "add_batch_hostfile"
        - "assemble"
        - "log_used_files"
        - "_write_finalized_config"
        - "copy_files_to_thisrun"
        - "modify_namelists"
        - "modify_files"
        - "copy_files_to_work"
        - "write_simple_runscript"
        - "report_missing_files"
        - "database_entry"
        - "submit"
```

**Figure 3.** Section if *esm_runscripts.yaml* defining the order of execution of python functions in a compute job. The user can easily add new (own) functions.

### 2.2.2   YAML Configurations

The configuration files, written in YAML syntax, are used for storing all known information, including on model components and coupled setups, HPC systems and batch systems, usage of external software, configuration of the ESM-Tools themselves, etc.. Or to put in another way: The python code is entirely oblivious of models, couplings, coupled setups, experiments and so on. Fig. (4) shows a section of the YAML configuration file for *fesom-1.4*, containing metadata on the model itself, which is parsed automatically by a sphinx-project to assemble the documentation. This should just serve as an example to show how effective this approach is. Not only can each piece of information be used to control the compilation and execution of the models, but also to a generate complete description of an experiment (and in this sense enable provenance tracking and reproducibility), or a how-to "cookbook" for certain models. Even the flow of execution of *esm_runscripts* itself is controlled by a YAML file instead of normal python code.

```
metadata:
        Institute: Alfred Wegener Institute
        Description:
                Multiresolution sea ice-ocean model that solves the equations
                of motion on unestructured meshes
        Authors: Dmitry Sidorenko (Dmitry.Sidorenko@awi.de), Nikolay V. Koldunov (nikolay.koldunov@awi.de)
        Publications:
                - "The Finite-volumE Sea ice-Ocean Model (FESOM2) <https://doi.org/10.5194/gmd-10-765-2017>"
                - "Scalability and some optimization of the Finite-volumE Sea ice-Ocean Model, Version 2.0 (FESOM2) <https://doi.org/10.5194/gmd-12-399
1-2019>"
```

**Figure 4.** Section of the *fesom-1.4* YAML configuration, containing the model metadata used to automatically assemble parts of the ESM-Tools documentation.

The YAML format is powerful yet easy to edit and understand by less experienced users. An extra parser was developed to correctly parse the model configuration files. This parser includes features such as (i) using only part of the parsed file (required blocks), (ii) inclusion of variable expansion, (iii) list expansion and (iv) mathematical expressions. The ability to read parts of a file (block) is useful when the modeller needs to switch off a component of the coupled system (e.g., biogeochemistry or ocean). Therefore, the same file can be used to implement an ESM with different combinations of model components. List expansion is not required, but helpful in shortening the YAML configuration files. For example, if the coupled fields are not known by default but are rather defined by the user, list expansion can be used to define the names of output files of the model. Mathematical expression such as calculating the previous or next year of a run is required when preparing the input files for a consecutive run or locating restart files.

Overall, the ESM-Tools functions and configuration files fulfill the following criteria:

**User-friendly:** The functions and configuration files are easy to understand by the users, and equally easy to edit and extend. This is mainly because the vast majority of the users don't need to be able to write Python code at all, as it is sufficient to work on the YAML configurations. Only for implementing new functionality of the ESM-Tools a modeller needs to write Python functions, but even then, our self-written plugin manager can be used to work on new functionality without touching the ESM-Tools core.

**Modular:** The ESM-Tools software is modular itself, in more than one way. First, we treat model components independently, and in the way we would treat stand-alone models. For a coupled setup as AWI-CM, that means we download each of the three components (atmosphere ECHAM, ocean FESOM and coupler OASIS) – if possible – from its own repository, then configure and compile it on its own. It is easy to change the url of the repository or the branch to be compiled in order to not stand in the way of users who make changes in the code, or who work on private development branches. In case of coupled setups we activate the coupling within the models by defining compile time switches. This of course means that the coupling itself, both technically by implementing calls to a dedicated coupling software, as well as physically by adapting the parametrization of the components, needs to be established beforehand. This is a time-consuming task, and the ESM-Tools should not be mistaken as a tool to couple two components - it is a tool to provide configurations for model components as well as existing coupled setups to scientists, and to make the choice of components in an ESM as easy as if it were Plug-and-Play.

Using the same model repository for a component, for coupled setups and stand-alone applications is a huge improvement for model developers who can more easily share their work with each other; it requires agreements and standards though that model developers follow in order to keep a code repository stable. An example of what *esm_runscripts* is doing exactly in the case of installing *awicm-2.0* is shown in fig. 5. Even though this is the strategy preferred by us and most active ESM-Tools users at the time being, it would also work to include whole coupled setups without referring to their components in a modular way.

```
[a270058@mlogin108% esm_master install-awicm-2.0
    Executing commands in this order:
        mkdir -p awicm-2.0
        cd awicm-2.0
        git clone -b 2.8 https://a270058@gitlab.dkrz.de/modular_esm/oasis3-mct.git oasis
        git clone -b 6.3.04p1 https://a270058@gitlab.dkrz.de/modular_esm/echam6.git echam-6.3.04p1
        git clone -b 2.0.2 https://a270058@gitlab.dkrz.de/FESOM/fesom2.git fesom-2.0
        sed -i '/set(FESOM_COUPLED/s/OFF/ON/g' fesom-2.0/CMakeLists.txt
        sed -i '/ECHAM6_COUPLED/s/OFF/ON/g' echam-6.3.04p1/CMakeLists.txt
        cd oasis
        mkdir -p build; cd build; cmake ..;   make -j 1; mkdir -p ../include/; cp lib/psmile/libpsmile.a lib/psmile/mct/libmct.a lib/psmile/mct/mpeu/l
ibmpeu.a lib/psmile/scrip/libscrip.a ../lib; cp lib/psmile/mod_oasis*mod ../include
        cd ..
        mkdir -p ./lib
        cp  oasis/build/lib/psmile/libpsmile.a  lib
        cp  oasis/build/lib/psmile/mct/libmct.a  lib
        cp  oasis/build/lib/psmile/mct/mpeu/libmpeu.a  lib
        cp  oasis/build/lib/psmile/scrip/libscrip.a  lib
        cd echam-6.3.04p1
        mkdir -p build; cd build; cmake ..;   make install -j `nproc --all`
        cd ..
        mkdir -p ./bin
        cp  echam-6.3.04p1/src/echam/bin/echam6  bin
        cd fesom-2.0
        mkdir -p build; cd build; cmake ..;   make install -j `nproc --all`
        cd ..
        mkdir -p ./bin
        cp  fesom-2.0/bin/fesom.x  bin
        cd ..
```

**Figure 5.** Output of *esm_master install-awicm-2.0*, listing the steps performed by *esm_master* to download, configure and compile the three components *echam-6.3.04p1*, *fesom-2* and *oasis3mct*. Note that preprocessor flags are used to change the model components from the standalone to the coupled configurations.

In the same way, we write the configuration files for the runtime infrastructure as if each component would be used as a stand-alone model, and just add a short configuration for each coupled setup, containing only the things that need to be changed for a component to be coupled. In this way it becomes very easy to add new components and couplings.

The second level of modularity we take special care of is the separation of information and functionality, meaning that we have the YAML configuration files containing the information we have (on models, setups, HPC systems, version control system,...), and Python functions for the functionality of the ESM-Tools. This implies that users / model developers who want to add e.g. new model components, scenarios, and meshes, don't need to write code at all, but merely edit nicely readable YAML lists.

**Portable:** The tools can run on different computer architectures. Furthermore, since the ESM-Tools is delivered with external packages, installation of external libraries is not necessary, making their installation easier.

**Maintainable:** The functions are short and simple enough for efficient maintenance by the developers.

**Extendable:** Since the system is modular, new ESMs or functions can be implemented in the tools fairly easy. For a new model component, coupled ESM, HPC system etc. a user needs to write an additional YAML configuration. New functionality can be added by writing Python functions, and using the plugin manager to schedule the execution of the function. In this way even developers who want to work on the ESM-Tools themselves do not have to read and edit the ESM-Tools core, which on one side makes it easier for the developer, but also for us as we can maintain the core functions independent of each other.

**Performance:** Using ESM-Tools is not detrimental to model performance, including high-resolution model configurations.

## 2.3 Structure and Architecture

### 2.3.1 Compile Time

The executable *esm_master* is a tool to download, configure and compile a variety of stand-alone models and coupled systems on different HPC systems. The command line utility provides a user-friendly interface to the underlying build environment that is provided by each model component or coupled setup. This enables the modeller to use the same syntax to install different models despite their difference in the underlying build environment, which remains unchanged as *esm_master* rather maps the commands required for each build to a common syntax. Figure 6 shows the schematic layout of the *esm_master* functioning. In this figure, models 1–3 have different building strategies (e.g. cmake, custom-made Makefile, autotools). However, the user is able to configure/compile the model without the knowledge of all the different building strategies or environments used by the different models. This results in effectively reducing the time required by the modeller to obtain a first model executable. Again, we want to point out that the main aim of the ESM-Tools is to make available easily and in a unified way what already exists, and that includes optimal HPC settings for compilation and execution of ESMs. In this sense, we collect the solutions that were already found by model developers, and make sure that they become the standard setting for the given model. In principle, that configuration could be very different for each model - it turns out though that with few exceptions, most of the coupled setups and standalone models supported by the ESM-Tools run well with the same settings. And in this way we again can help the model developers to get an idea how to utilize an unknown model or HPC system by providing an environment setting that is known to work well for other users.

### 2.3.2 Run Time

The *esm_runscripts* tool is the most extensive component of the ESM-Tools in terms of functionality. This tool consists of optimized implementations of all the functionalities to set up all required phases to run an ESM simulation. The user only needs to provide a short script that includes experiment specific definitions, while the *esm_runscripts* execute all the phases of a simulation in the correct order (see Figure 7).

The first task of this tool is to prepare the simulation setup. This includes creating the folder structure of the run, allocating the run files/folders, preparing/allocating initial and forcing files and preparing the model input files via the esm_parser package (see Section 2.2). The next task is to run the model by submitting the executable to an HPC batch system or job scheduler.

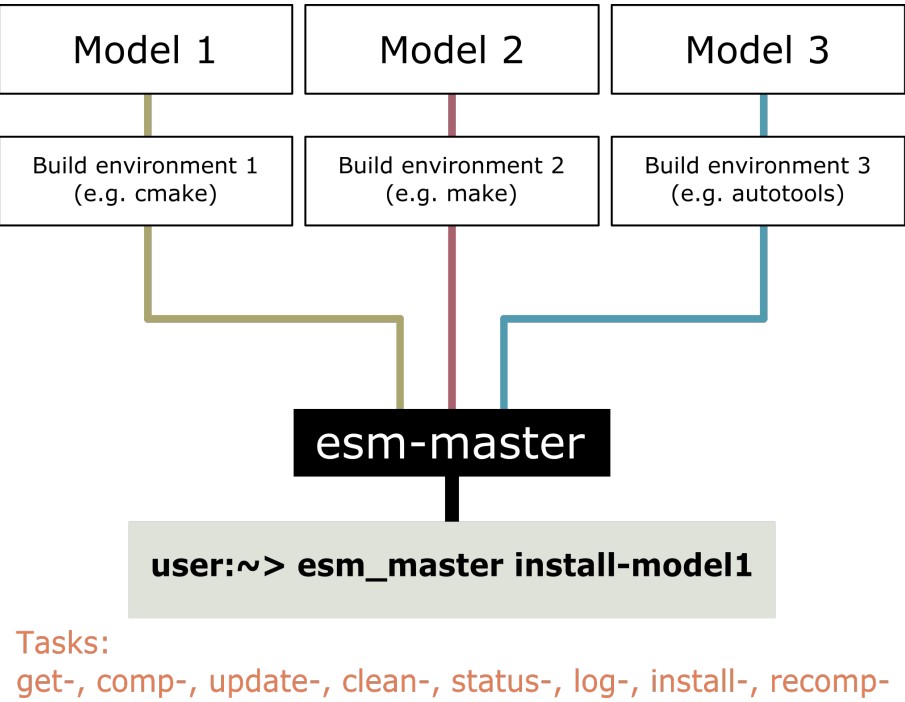

**Figure 6.** Schematic of the *esm_master* tool, which provides generic building commands that map to each model component specific building environment. Further generic tasks are: comp-, update-, clean-, status-, log-, install-, recomp-.

During the execution of the run, the tool monitors the simulation, creates log files and provides feedback on the progress of the run. After the completion of the run the log files and output data are moved to designated folders. Finally the post-processing of the output data, including preparation of the consequent run is carried out.

### 2.3.3 Online Monitoring

    The *esm_viz* is a command line tool to schedule automatic monitoring of Earth system model simulations. A key benefit here is
the ability to visualize particular aspects of a simulation as it progresses, since particularly for long-running jobs waiting for the job to complete (which may take several weeks) before noticing any errors can be avoided. As with the remainder of the ESM-Tools, *esm_viz* is configured with a YAML file. Plots representing climatological averages and timeseries can be automatically generated, which are interactively displayed in the browser and can be downloaded in a high-quality format for inclusion in other scientific outputs, such as conference posters, papers, or talks. Additionally, *esm_viz* gives you an overview of your job
in relation to the remainder of the supercomputer, with an estimated completion time, approximations of queuing time and real model-throughput, and links to relevant log files. Notably, *esm_viz* runs on a separate computer to the supercomputer, and is scheduled via a cronjob, allowing for independent monitoring and reporting even if the main computer running the simulation

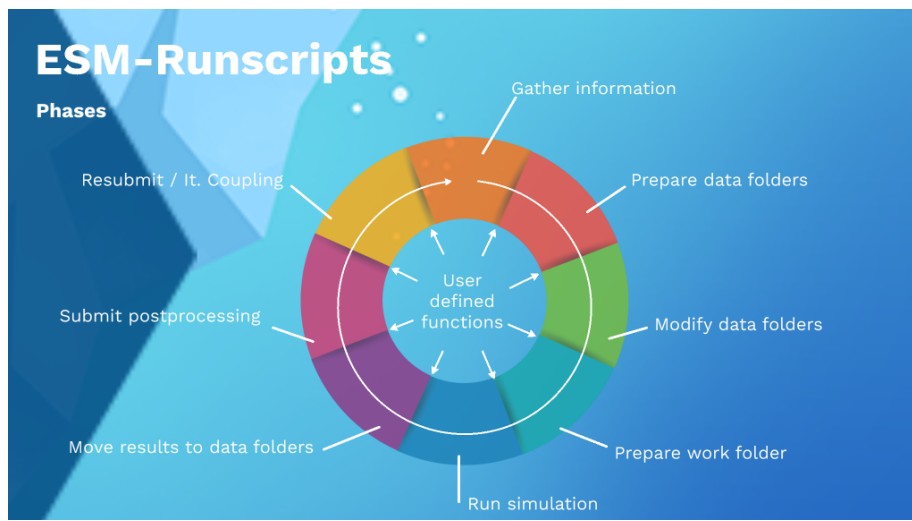

**Figure 7.** Schematic of the tasks carried out by *esm_runscripts*. In addition to standard job phases, users and developers have the possibility to add own python code via a plugin manager, which can be executed between the pre-defined function calls without the need to edit ESM-Tools core functionality (denoted as "User defined functions").

goes offline. In this case, *esm_viz* also provides information about the last available state of the simulation. For all further information we refer the reader to a future publication.

### 2.3.4 Graphical User Interface

A prototype Graphical User Interface (GUI) was developed for a previous version (v2.0) of the ESM-Tools (see also Figure 8) and is, to our knowledge, the first developed GUI that incorporates multiple ESMs. The development of the GUI was possible due to the modular layout of the repositories of the models used by the ESM-Tools, as well as treating each component separately in what has become the YAML configurations. Users are able to download and compile the chosen setup or models, which are selected from standard or customer config tab.

### 2.4 Supported Models, Coupled Systems and HPC Environments

The current version of the ESM-Tools contains configuration files for four ocean models, three atmosphere models, two bio-geochemistry models, an ice sheet model, an isostatic adjustment model, a hydrology model, and a land-surface model (see Table 2 for details). It must be clear that using the ESM-Tools, which are under GPLv2 OpenSource License themselves, does not include licences to these models, and the download tool *esm_master* will only work if the corresponding paths are set to valid repositories that the user actually has access to.

Two different batch systems are currently supported by the tools: Slurm[8] and Moab[9].

---

[8]https://slurm.schedmd.com/documentation.html
[9]http://docs.adaptivecomputing.com/mwm/7-1-3/help.htm

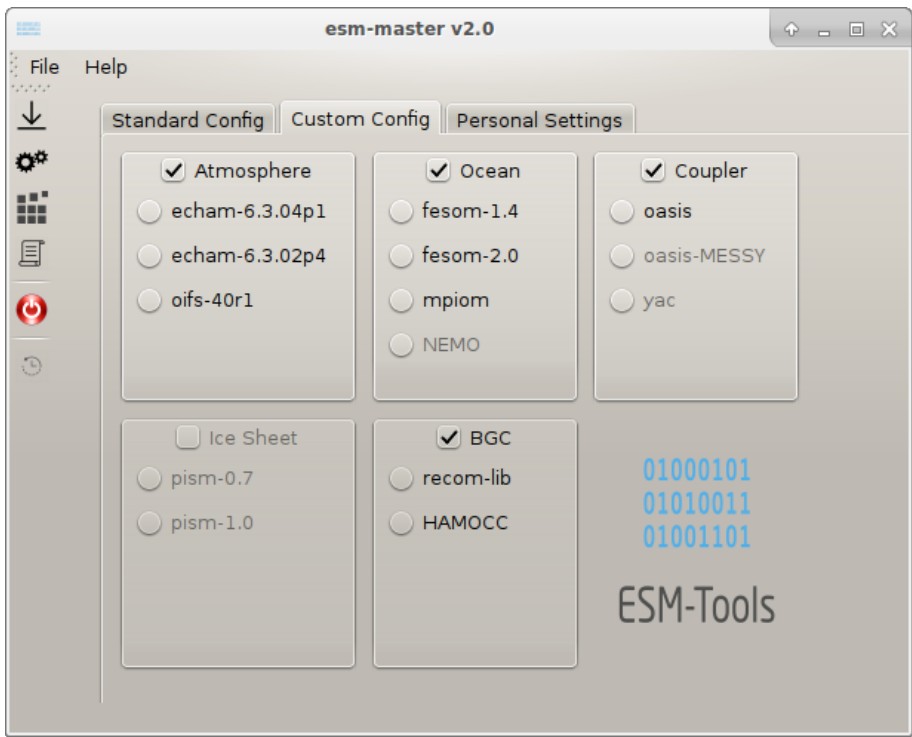

**Figure 8.** GUI for ESM-Tools, which helps to download, compile, and configure selected models.

The ESM-Tools, along with the available model configurations, can be readily run on several German HPC centers, such as the Norddeutsche Verbund für Hoch- und Höchstleistungsrechnen (HLRN), the German Climate Computing Center (DKRZ), the Jülich Supercomputing Centre (JSC) and the Alfred-Wegener Institute supercomputer. The run systems and modules on these machines are regularly updated or even completely changed. Therefore, the relevant ESM-Tools machine files are also adapted accordingly by the ESM-Tools support team. This leads to the modellers saving a considerable amount of time needed for understanding the new system and adapting their files.

## 3 ESM-Tools Development

### 3.1 Software and Code Management

The first two versions of the ESM-Tools were released in September 2018 and March 2019, respectively. In these versions, the functions were mainly coded in bash, and code was mixed with model configuration. With the growth of the tools and the addition of various ESMs, maintaining the functions written in bash became non-trivial and time consuming. This was mainly due to the bash language not being object-oriented in nature. Therefore, in the third version of the tools, a complete migration of the functions to Python was carried out. The ESM-Tools software is currently hosted and developed on GitHub (see also

| Model component | Model type | Reference |
|---|---|---|
| FESOM 1.4 | Ocean | Danilov et al. (2004) |
| FESOM 2.0 | Ocean | Danilov et al. (2017) |
| MPIOM | Ocean | Jungclaus et al. (2013) |
| NEMO | Ocean | Madec et al. (2015) |
| ECHAM | Atmosphere | Stevens et al. (2013) |
| ICON-A | Atmosphere | Giorgetta et al. (2018) |
| OpenIFS | Atmsophere | https://www.ecmwf.int/en/research/projects/openifs |
| ReCom | Biogeochemistry | Hauck et al. (2013); Schourup-Kristensen et al. (2018) |
| PISM | Ice sheet | Winkelmann et al. (2011) |
| Vilma | Isostatic adjustment | Martinec (2000) |
| HD-model | Hydrology | Hagemann and Dümenil (1998) |
| JSBach | Land-surface | Raddatz et al. (2007) |

**Table 2.** Supported model components

| Coupled Setup | Model Components | Coupler | Reference |
|---|---|---|---|
| AWI-CM-v1.0 | FESOM-1.4, ECHAM-6.3 | OASIS3-MCT | (Sidorenko et al., 2015; Rackow et al., 2018) |
| AWI-CM-v2.0 | FESOM-2.0, ECHAM-6.3 | OASIS3-MCT | (Sidorenko et al., 2019) |
| AWI-CM-RECOM | AWI-CM-v2.0, REcoM2 | OASIS3-MCT | (in prep.) |
| AWI-CM-PISM | AWI-CM-v2.0, PISM | OASIS3-MCT, SCOPE? | (Gierz et al., in prep.) |
| OpenIFS-FESOM | FESOM-2.0, OpenIFS 43 | OASIS3-MCT | (Streffing et al., in prep.) |
| OpenIFS-NEMO | NEMO-3.6, OpenIFS 43 | OASIS3-MCT | (in prep.) |
| FOCI | NEMO-3.6, ECHAM-6.3 | OASIS3-MCT | (Matthes et al., in prep.) |
| MPI-ESM | MPIOM, ECHAM-6.3 | OASIS3-MCT | (Giorgetta et al., 2013) |
| MPI-ESM-PISM | MPI-ESM, PISM | OASIS3-MCT | (Ziemen et al., 2019) |

**Table 3.** Supported coupled setups

*Code availability*). The ESM-Tools core development team consists of four scientists and scientific programmers from two groups (Climate Dynamics and Paleoclimate Dynamics) at the Alfred-Wegener Institute (AWI). These developers take care of the strategic planning and implementation of new main features, as well as organizing regular developer meetings, workshops for new users, etc.. On a second tier, more than 30 scientists from AWI, but also from GEOMAR Helmholtz-Zentrum für

Ozeanforschung Kiel and Helmholtz-Zentrum Potsdam Deutsches GeoForschungsZentrum (GFZ), authored contributions to the Tools, which makes it an active OpenSource community.

Special focus has been placed upon the documentation, for which we try to provide everything that is possible - from a standard user manual, command line help, man pages etc. up to a website (https://www.esm-tools.net) containing course material, and even a YouTube channel (*ESMTools*) with tutorial videos.

To efficiently organize the development of the model, agile software development is applied (Dingsøyr et al., 2012). Since software requirements in scientific projects are often unclear at the start (Wieters and Fritzsch, 2018), agile software development allows for the adaption of the software to the user requirements during the development. Regular weekly and monthly meetings are organized to manage the development. During the weekly meetings, the lead development team discusses the progress of the tools and plans the timeline for the next week. During monthly meetings, the larger development team, includ-

ing members of other groups from partner institutes, discusses the status of the development, issues and problems, and newly implemented features. Furthermore, requirements for future developments, collected from users, are also reviewed. In this way, the road map for future releases is continuously adapted to meet the target of two software releases per year.

## 3.2 Contributing to the Development

The development of the ESM-Tools consists, to a large extent, of the contribution of users who develop their own extensions

to the existing code. To extend the list of supported models, coupled systems and HPC systems (see also Section 2.4), users are encouraged to develop new features and functionalities according to their requirements. This can be done by contributing to the development on GitHub (e.g., by pull requests). Furthermore, users can suggest a "wish-list" of missing functionalities (e.g., via issue tracker). The ESM-Tools developers then work in close collaboration with the users to implement and test the new features. Regular workshops are also organized for users and developers. In addition to the development, the most

important contribution by users is reporting model errors and bugs, which are then resolved by the core developers. Apart from adding to the python code for more ESM-Tools core functionality, the main contribution users have provided in the past was the addition of additional model components, coupled setups or experiment parameters into the various YAML configuration files. Typically, to include an additional model component, the following steps should be followed:

- – Make your code available on a *git* or *svn* server. That should be a standard anyway.

– Create a pool directory on the server you want to work on, containing the needed forcing, input and restart data.

- – Add standard namelists / configuration files to *esm_tools*. These can be templates changed by the tools later on.

- – Create a model YAML configration file, containing the information about how to download and compile the model. An an example, in fig. (9), we show the entries controlling download and compilation for *fesom-1.4*. This should enable all the functionality of *esm_master*.

– Add the information about needed files, e.g. forcing data, naming of restart files, namelists etc..

```
# FESOM YAML CONFIGURATION FILE
#

model: fesom
branch: master
version: "1.4"
type: ocean

available_versions:
- 1.4-esm_interface
- '1.4'
- 1.4-recom-awicm
choose_version:
  '1.4-recom-awicm':
    destination: fesom-1.4
    branch: co2_coupling
  '1.4-esm_interface':
    destination: fesom-1.4
    branch: using_esm-interface

git-repository: https://gitlab.dkrz.de/modular_esm/fesom-1.4.git
install_bins: bin/fesom
clean_command: ${defaults.clean_command}
comp_command: mkdir -p build; cd build; cmake ..;   make install
```

**Figure 9.** Excerpt from *fesom.yaml* with the information on how to compile and download *fesom-1.4*.

```
namelist_changes:
        namelist.config:
                clockinit:
                        yearnew: "${initial_date!syear}"
                calendar:
                        include_fleapyear: "${leapyear}"
                paths:
                        ForcingDataPath: "${forcing_data_dir}"
                        MeshPath: "${mesh_dir}"
                        OpbndPath: "${opbnd_dir}"
                        ClimateDataPath: "${climate_data_dir}"
                        TideForcingPath: "${tide_forcing_dir}"
                        ResultPath: "${work_dir}"
                timestep:
                        step_per_day: "${steps_per_day}"
                        run_length: "${restart_rate}"
                        run_length_unit: "${restart_unit}"
                inout:
                        restartflag: "${restart_flag}"
                        output_length: "${restart_rate}"
                        output_length_unit: "${restart_unit}"
                        restart_length: "remove_from_namelist"
                        restart_length_unit: "remove_from_namelist"
                mesh_def:
                        part_format: "remove_from_namelist"
```

**Figure 10.** Excerpt from *fesom.yaml*, specifying how to change the namelist templates during the preparation phase of a run.

– Add at least the parameters for one standard experiment for testing, including namelist changes etc.. In fig. (10) you can see the an example. again from *fesom-1.4*.

– Add a sample runscript for this experiment, preferably in YAML format.

As this is just a short overview and a detailed how-to section is beyond the scope of this paper, we kindly invite the interested reader to have a look on our quite extensive documentation, both on github as well as our webpage, or use the existing configuration files as a source of inspiration.

## 4 Applications

Currently around 75 scientists at AWI and partner institutes are registered users of the ESM-Tools. This includes mostly model users, but also model developers / maintainers, and system administrators. The tools are actively used to run the AWI Climate Model (AWI-CM). In fact, all contributions of AWI to the Coupled Model Inter-comparison Project, CMIP6, were produced using the ESM-Tools. The ESM-Tools are also used to run paleo-climate simulations. The application of the tools has resulted in a more efficient exchange of model configurations and input and output model data between modellers and scientists.

## 5 Discussion

ESM-Tools provides a standardised way of working with ESMs. In addition to the benefits of applying ESM-Tools mentioned above, the development of ESM-Tools has led to various new insights. From a technical perspective, using an object-oriented language for such a software tool has proven to be essential for efficient expansion and maintenance of the tools themselves. The previous versions of the ESM-Tools, written in bash, suffered from lengthy functions as well as entangled functions and configuration files that were hard to maintain. The separation of configuration files and ESM-Tools functions in v3.0 as well as using a high-level language has made ESM-Tools more modular and easy to maintain.

From a community-building perspective, the development has brought together scientists and modellers from various backgrounds and institutes, resulting in a new community of modellers and scientific programmers in the geosciences. Through developing and using ESM-Tools, programmers and modellers have efficiently shared their knowledge on best coding practices and learned to collaborate with each other more closely. In conclusion, ESM-Tools has developed into an active OpenSource community, making it much easier for modellers, model developers and system administrators to obtain/provide access to, run and maintain Earth system models in a standardized way.

Apart from the things ESM-Tools can do to simplify model usage for the modellers, and that were discussed already extensively in this paper, it is also important to point out what the ESM-Tools are **not**:

- A coupling software (like OASIS or YAC)

- A tool that automatically couples model components – even though our tools, especially the GUI, look as if we can couple model components in a "plug-and-play" way, that is of course not the case. Coupling is a complex and time consuming task, and behind each of the couplings supported in the ESM-Tools, several person months of technical work, parametrization and fine tuning is hidden. The ESM-Tools only make the already established couplings as easy available as if it was plug-and-play.

- A replacement for model build systems / model runscripts – we understand our tools as alternatives, not replacements, meaning trying the ESM-Tools doesn't influence already existing solutions, and the user can switch back to them at any time.

- A coding standard / collection of generic interfaces / DSL. Using the ESM-Tools doesn't involve changes to the model codes at all, we provide an external infrastructure.

Generic interfaces are the main purpose of a second software, called *esm-interface*, that is currently under development. It provides a library of generic procedures to enable a modular approach to ESM coupling. This software will be discussed in a separate publication (in preparation).

In addition to the benefits of applying ESM-Tools mentioned above, the development of ESM-Tools has led to various new insights. From a technical perspective, using an object-oriented language for such a software tool has proven to be essential for efficient expansion and maintenance of the tools themselves. The previous versions of the ESM-Tools, written in bash, suffered from lengthy functions as well as entangled functions and configuration files that were hard to maintain. The separation of configuration files and ESM-Tools functions in v3.0 as well as using a high-level language has made ESM-Tools more modular and easy to maintain.

From a community-building perspective, the development has brought together scientists and modellers from various backgrounds and institutes, resulting in a new community of modellers and scientific programmers in the geosciences. Through developing and using ESM-Tools, programmers and modellers have efficiently shared their knowledge on best coding practices and learned to collaborate with each other more closely. In conclusion, ESM-Tools has developed into an active OpenSource community, making it much easier for modellers, model developers and system administrators to obtain/provide access to, run and maintain Earth system models in a standardized way.

*Code availability.* ESM-Tools is OpenSource software developed at the Alfred Wegener Institute Helmholtz Centre for Polar and Marine Research and licensed under a modified GPLv2 licence currently hosted on https://github.com/esm-tools/esm_tools.git, with a mirror on https://gitlab.awi.de/esm_tools/esm_tools.git. Please contact the main developers (dirk.barbi@awi.de, nadine.wieters@awi.de) for development access. Code's DOI: 10.5281/zenodo.3737928.

## Appendix A: Supporting Information and Documentation

The ESM-Tools is well documented and therefore easy to install and understand. The documents are constantly updated by the model developers. Users are also encouraged to contribute to the documentation.

In order to facilitate the use of ESM-Tools, regular workshops are conducted for users and developers. The users can also report bugs/problems via gitlab, which are then solved by developers. The ESM-Tools Webpage: https://www.esm-tools.net/ provides essential information on the ESM-Tools. As an outreach, monthly newsletters containing report on new features, bug

fixes and up coming events regarding the ESM-Tools is released. The users can also refer to Twitter: @ToolsEsm #ToolsEsm

for further updates.

## A1  Example ESM-Tools runscript

```
set -e

setup_name="awicm"
#check=1

account=ab0995
compute_time="00:15:00"
##########################################################################

INITIAL_DATE_awicm=2000-01-01        # Initial exp. date
FINAL_DATE_awicm=2000-02-01          # Final date of the experiment

awicm_VERSION="CMIP6"
POST_PROCESSING_awicm=0
SCENARIO_awicm="PI-CTRL"

RES_fesom=CORE2

MODEL_DIR_awicm=${HOME}/esm_yaml/awicm-CMIP6/
BASE_DIR=/work/ollie/dbarbi/esm_yaml_test/

POOL_DIR_fesom=/work/ollie/pool/FESOM/
MESH_DIR_fesom=/work/ollie/pool/FESOM/meshes_default/core/

NYEAR_awicm=0            # Number of years per run
NMONTH_awicm=1           # Number of months per run

LRESUME_echam=0
LRESUME_fesom=0
LRESUME_oasis3mct=0

RESTART_RATE_fesom=1
RESTART_FIRST_fesom=1
RESTART_UNIT_fesom='m'

further_reading_fesom="fesom_output_control.yaml"

##########################################################################
load_all_functions
general_do_it_all $@
```

**Figure A1.** Example runscript, coupled model setup AWICM-2.0, shell style

## A2  Example YAML configuration file

```
################################################################################
######################## AWICM 1 YAML CONFIGURATION FILE #######################
################################################################################

general:
        model: awicm
        model_dir: ${esm_master_dir}/awicm-${version}

        coupled_setup: True

        include_models:
                - echam
                - fesom
                - oasis3mct

        version: "1.1"
        scenario: "PI-CTRL"
        resolution: ${echam.resolution}_${fesom.resolution}
        postprocessing: false
        post_time: "00:05:00"
        choose_general.resolution:
                T63_CORE2:
                        compute_time: "02:00:00"
                T63_REF87K:
                        compute_time: "02:00:00"
                T63_REF:
                        compute_time: "02:00:00"

################################################################################
########### necessary changes to submodels compared to standalone setups #######
################################################################################

echam:
        restart_firstlast: "first"
        namelist_changes:
                namelist.echam:
                        runctl:
                                lcouple: .true.
        adj_input_dir: "${fesom.mesh_dir}/tarfiles${echam.resolution}/input/echam6"
        model_dir: ${general.model_dir}/echam-${echam.version}
        setup_dir: ${general.model_dir}
        ocean_resolution: "${fesom.resolution}"
        remove_forcing_files:
                - sst
                - sic
        version: "6.3.04p1"

        choose_general.resolution:
                T63_CORE2:
                        nproca: 24
                        nprocb: 18
                T63_REF87K:
                        nproca: 24
                        nprocb: 18
                T63_REF:
                        nproca: 24
                        nprocb: 18

################################################################################

jsbach:
        #dynveg_file_ending: ""
        adj_input_dir: "${fesom.mesh_dir}/tarfiles${echam.resolution}/input/jsbach"
        namelist_changes:
                namelist.jsbach:
                        hydrology_ctl:
                                gethd: "remove_from_namelist"
                                puthd: "remove_from_namelist"
        version: "3.20"
```

**Figure A2.** Example of YAML configuration file, coupled model setup AWICM-2.0, part 1/2

```yaml
fesom:
        choose_general.version:
                1.1:
                        version: "1.4"
                CMIP6:
                        version: "1.4"
                2.0:
                        version: "2.0"
        choose_general.resolution:
                T63_CORE2:
                        nproc: 288
                T63_REF87K:
                        nproc: 216
                T63_REF:
                        nproc: 128

        opbnd_dir: ""
        tide_forcing_dir: ""
        forcing_data_dir: ""
        model_dir: ${general.model_dir}/fesom-${fesom.version}
        setup_dir: ${general.model_dir}

        add_namelist_changes:
                namelist.oce:
                        boundary:
                                restore_s_surf: 0.0

#####################################################################################

oasis3mct:
        model_dir: ${general.model_dir}/oasis

        process_ordering:
                - fesom
                - echam

        a2o_lag: "${echam.time_step}"
        o2a_lag: "${fesom.time_step}"
        a2o_seq: 2

        coupling_time_step: 3600
        coupling_target_fields:
                o2a_flux:
                        - 'sst_atmo:sit_atmo:sie_atmo <--distwgt-- sst_feom:sit_feom:sie_feom'
                        - 'snt_atmo <--distwgt-- snt_feom'

                a2o_flux:
                        - 'taux_oce:tauy_oce:taux_ico:tauy_ico <--bicubic-- taux_atm:tauy_atm:taux_ica:tauy_ica'
                        - 'prec_oce <--distwgt-- prec_atm'
                        - 'snow_oce <--distwgt-- snow_atm'
                        - 'evap_oce <--distwgt-- evap_atm'
                        - 'subl_oce <--distwgt-- subl_atm'
                        - 'heat_oce <--distwgt-- heat_atm'
                        - 'heat_ico <--distwgt-- heat_ica'
                        - 'heat_swo <--distwgt-- heat_swa'
                        - 'hydr_oce <--distwgt-- hydr_atm'

        coupling_directions:
                'feom->atmo':
                        lag: ${o2a_lag}
                        seq: 2
                'atmo->feom':
                        lag: ${a2o_lag}
                        seq: ${a2o_seq}

        coupling_methods:
                distwgt:
                        name: distwgt
                        bins: 15
                        other_number: 6
                bicubic:
                        name: bicubic
                        bins: 15
```

**Figure A3.** Example of YAML configuration file, coupled model setup AWICM-2.0, part 2/2

*Author contributions.* D. Barbi is the lead developer of ESM-Tools; N. Wieters is supporting developer of ESM-Tools and contributed to the development of parts of them, documentation, branches management and user support; P. Gierz is supporting developer of ESM-Tools and

415  contributed to the development of parts of them, documentation, Python implementation, branches management and user support; F. Chegini contributed to software development in python, user support, and documentation management; S. Khosravi developed the GUI's first version; L. Cristini contributed to team management and software project management.

*Competing interests.* The authors declare that they have no conflict of interest.

*Acknowledgements.* We especially want to thank Thomas Jung and Jan Streffing (AWI), Joakim Skjellson and Sebastian Wahl (GEOMAR)

420  for major contributions and important feedback, Jaroslav Piwonski (University of Kiel) for helping us to organize the reimplementation in Python / YAML, and Tido Semmler, Christopher Danek and Christian Stepanek (all at AWI) for lots of testing and bug reporting. The work described in this paper has received funding from the Initiative and Networking Fund of the Helmholtz Association through the project "Advanced Earth System Modelling Capacity (ESM).

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
