# Peer review of "ESM-Tools Version 5.0: A modular infrastructure for stand-alone and coupled Earth System Modelling (ESM)"

_Geoscientific Model Development, 2020_

## Referee Comment (RC1) · Anonymous Referee #1 · 20 Sep 2020

**\*\*GENERAL COMMENTS\*\***

The manuscript by Barbi et al. presents the ESM-Tools v4.0, which is a modular, open-source software to "use" Earth System Models more efficiently, by providing a standardised infrastructure that covers, orchestrate, and wraps around all aspects of working with ESMs from obtaining the model's source code, building on different HPC systems, run control, monitoring, to post-processing.

This topic is highly applied and very relevant for the geoscience modelling community, it is in the scope of GMD and appropriate, presenting novel tools for, very broadly speaking, model system maintenance. The software is well designed and follows a

logic and concise concept. It is well maintained as community code on github.

Overall, however, in the opinion of the reviewer, the manuscript in its current form needs a careful restructuring within the individual sections and many refinements throughout in order to help the reader fully appreciate and understand the capability and potential of the ESM-Tools. Many parts lack a framing into the bigger overall context or even the ESM-Tools themselves. Also, some technical aspects should be presented in more detail to allow the readers to decide whether investing time in testing and test-implementing makes sense for their respective situation.

Apart from this general criticism, the manuscript by and large reads well and fluent. The Abstract and Appendix are OK, but as the rest of the manuscript need reworking, see above and detailed comments below. A more detailed schematic on a ECM-Tools workflow could be aided by more examples in the appendix; or those examples provided should at least be integrated more tightly. Also, more references could be provided to provide more background information.

**SPECIFIC COMMENTS**

(P: page, L: line or lines)

1. While the referee agrees with most of what is stated in the introduction, the introduction is lacking references, e.g., on current developments of ESMs, in the context also of HPC development, which makes using coupled model systems especially complex, by making, e.g., full use of emerging heterogeneous HPC systems. References that point into this direction might be, e.g., Bauer et al. (2015, Nature, doi: 10.1038/nature14956), Schulthess et al. (2019, Computing in Science Engineering, doi:10.1109/MCSE.2018.2888788), or Fuhrer et al. (2018, GMD, doi:10.5194/gmd-11-1665-2018), to name a few.

2. The sentence on P2 L26ff implies that ESM-Tools in fact aids in implementation of the the coupling itself; however, the lacking modularity of the ESMs component models

and difficulty of coupling them is not solved for models which are not part of the ESM-Tools, which are not a generic tool to help in the coupling as such (later on this is also clearly pointed out). Perhaps rephrase these parts so the actual scope of the software, which indeed can substantially aid in setting up, maintaining, using coupled model systems, becomes more clear. Maybe it already helps by moving the ESM-Tools notion out of the first paragraphs and further to the end of the introduction and focus in the first two paragraphs of the introduction on characteristics of state-of-the-art ESMs and HPC systems and challenges in efficiently developing and implementing and maintaining ESMs (the paragraph on the workflow involved) as a state-of-the-art and definition of the problem. In this part of the manuscript, it might help to clearly identify and list the different challenges along the workflow. This would put the listing of similar tools such as the ESM-Tools, which comprises most of the remainder of the introduction, into a better perspective.

3. The term "provenance" is not mentioned although the ESM-Tools offer a very nice auto-documenting functionality by means to YAML files, which eventually describe the setup and configuration very nicely. With these features a provenance tracking of complex numerical model runs becomes possible and lays the foundation for "reproducibility".

4. The manuscript could better or earlier-on in the text elaborate, that a prerequisite of the ESM-Tools usage is, in case of a multi-model ESM, that either modified source code repositories are used or some sort of source code patching is done, in case official source codes are used, so that the specific model coupling procedures are taken care of. This related to comment Nr.2 above, the crux of ESM software modularity is (among other things) the coupling as such. This is mentioned very clearly in the Discussion, but the manuscript would be more clear if these aspect were more explicitly addressed beforehand, where the manuscript seems a bit imprecise.

Additional comments to different parts of the manuscript:

"Abstract" - P1 L4f: I agree that the number of coupled setups increases, but I cannot see why this leads to a growing number of available versions of component models, except it means that they all need to be equipped with the respective coupler technology. - P1 L5f: Specify "system-dependent", it is HPC-system dependent; also aside from obtaining and operating, it is also "building" and "configuring", as mentioned further on in the introduction. - The content of the footnote should be more emphasised by putting this information in the introduction after the reader can appreciate why a focus on coupled model system is more rewarding, perhaps at the end of the introduction when the focus and scope of the manuscript is given.

"Introduction" - P1 L20: Perhaps expand the list here, not only climates are studied but all kinds of interacting and related processes in the Earth system for a multitude of reasons - P1 L21: Also "hydrology" should be mentioned in the list and the list refers mainly to "compartments" - P2 L23: Is it the ESM that lacks modularity or rather the individual component models, which make up the ESM, and which have been developed for a specific purpose? - P2 L48: Mentioning DSLs, cite, e.g., Lawrence et al. (2018, GMD, doi: 10.5194/gmd-11-1799-201), see also the general comments. - P2 L44ff: In case tools like MESSy are mentioned, wouldn't then also tools such as the Earth System Modelling Framework (ESMF)? - P3 L83: Which "specific criteria" are you refering to exactly?

"ESM-Tools Description" - This is a central part of the manuscript; as mentioned above, a restructuring and adding more information might help in better understanding the functioning of the tools. - P4 L93ff: As the introduction only gives little systematic overview of the individual steps and tasks involved in eventually getting an ESM to run ("to use"), I feel this paragraph is too detailed and lacks some introduction. I would suggest to introduce the structure, the overarching concept, the design principles of the ESM-Tools first. For example in on P4 line 93: which functions (I assume the individual tools that comprise the ESM-Tools and their functionalities)? - P4 L98: Rephrase: this reads as if the reader has to know the previous versions of the ESM-Tools. - P4

L110: What can be done with the plugin manager in detail, which functionalities do the ESM-Tools authors envision? - P5 L113ff: As above, without prior knowledge of the ESM tools, this section appears on the one hand side very detailed and on the other hand side it lacks introductory information: for which aspects of the complete workflow do YAML files exist? - P6 L137ff: This section reads as if different component models can be integrated and coupled with ESM-Tools. But in fact only those component models and combinations thereof, for which a dedicated coupling has been implemented already, irrespective of the ESM-Tools, purely driven by the coupling paradigm and software used (e.g., YAC, OASIS-MCT, MESSy, ESMF, etc.), can be integrated into the ESM-Tools and "managed" and re-arranged in an efficient way. This is how the reviewer understands the ESM-Tools functionality. See also the general remarks on this issue above. This is not a shortcoming of the ESM-Tools but should perhaps just be more clearly and explicitly stated – as in the Discussion. - P7 Fig1: Instead of this schematic, perhaps provide a workflow overview, which shows which tools use which YAML files etc., maybe split into different aspects of model usage: retrieving source code, building, configuring, etc. Also the further esm-master options should be further explained if they are mentioned. - P8 Fig2: The figure might be replaced by a listing, or a figure more in coherent with the schematic of Fig.1 - P7 L165ff: As much as the reviewer understands that it makes sense to provide a fully integrated system which covers all aspects of the "use" of an ESM, in how far do ESM-Tools compare run time functionalities as covered by the esm_runscripts compare to tools such as ecFlow, cylc, or JUBE? Do the esm_runscripts offer a specific benefit? - P8 L175f: Why is esm_siz not mentioned in the overview table 1; again here: given the rather complex implementations of cron jobs etc. to run the esm_viz, a schematic showing the overall structure of the ESM-Tools or a workflow would be very helpful - P8 L185ff: Does the GUI automatically pick up which setups, configurations etc. are possible depending on the available YAML files as part of an ESM-Tools installation? - P9 L192ff: A crucial and important aspect seems to be where the model systems come from and how they are provided: Just retrieving any component model from its official repository without the extra very time-consuming step of coupling this component model to other compartment models does not help; i.e. the component models either have to be modified a-priori and are provided through specific repositories or official model codes are retrieved and then patched to provide the coupling functionality. This is the lacking ESM component model modularity, wich is mentioned in the introduction and also tackled in several model development approaches towards unified model systems with interchangeable components (e.g., the SIMA efforts by NCAR). - P9 L200: Perhaps more HPC information should be given as this seems a major problem in using ESMs, to efficiently set them up (i.e. compile, arrange data flow-paths, processing chains, etc. wich incorporate the intricacies of the respective HPC system) - P10 Tab3: I think this table should also include information on the coupler used, as this is determining how truly modular component models can be combined. - P11 Fig4: Given the Tab2 and Tab3, the reviewer cannot see how Fig.4 provides extra information. - The examples of the YAML files in the Appendix should be included more in the ESM-Tools Description. Good way of maintaining reference implementations.

"ESM-Tools Development" P11 L209: The ESM-Tools seem to follow various best practices in modern software development. Does a formalised software development plan exist? ESM-Tools are provided via github, this could also be mentioned here. Also: Can you provide any recommendations in software development, which are worth being shared?

"Discussion" - P13 L252ff: See my comments above, just to avoid confusion, this aspect should be mentioned earlier on. - P14 L295ff: This is a concise summary and despite the fact that it is mentioned that ESM-Tools provides a standardised way of working with ESMs it could be emphasised more before. - Perhaps information could be provided for interested modelling groups how long it takes approximately to get a new model system or a single component model integrated into the ESM-Tools.

**TECHNICAL CORRECTIONS**

- All text: Check acronyms and abbreviations. - L3: replace "...components of the Earth..." with "...compartments of the Earth..." - L131: oasis should be in capital letters, OASIS - L287: object-oriented

---

## Referee Comment (RC2) · Anonymous Referee #2 · 30 Nov 2020

**General comments**

Utilising Earth System Models (ESMs) for numerical experiments in climate research is undoubted a complex task for many scientists and any attempt to overcome the complexity is welcome. The ESM-Tools constitute such an attempt and the list of supported models, components, configurations, and research groups testifies that the approach has successfully used for a wide range of use cases. The article gives, in more than one respect, a good overview about the concept, implementation, and usage of the ESM-Tools. As such, the topic is very relevant and well within the scope of the GMD journal.

[Figure]

The ESM-Tools cover support for all common workflow steps for numerical climate experiments. The software appears well structured and build on clear concepts. The effort to document, provide information and generally interact with users via different channels is impressive.

The manuscript is well structured and covers the essential aspects of the ESM workflow, as well as a good overview about concepts and implementation. Nevertheless, revision is needed, in my opinion, with respect to the following aspects:

(1) General overview of the overall workflow

It should, throughout the text, be made clearer how the ESM-Tools work in detail. Examples should be given in terms of specific ESM-Tool commands for workflows such as:

* getting an existing model to run from scratch

* adapt a new model to the ESM-Tools

* add a new function/feature

The examples could be given in the form of YAML snippets (see next point) to better illustrate the workflow description.

(2) Examples or YAML snippets

The ESM-Tools are controlled by YAML files and as such a crucial concept, it needs a clear description. The YAML files in the appendix are good, but it would help the understanding to give short YAML snippets in the text to illustrate the features explained. A particular example is section 2.1.2 where extensions to YAML are listed.

(3) Language

The language of the manuscript is sometimes rather informal or unprecise, such as in the following examples:

[Figure]

* p2l37 "often unnecessary"

* p4l107: "as many as sensible"

* p14l283: "cost the nerves"

These phrases leave sometimes the impression of opinions, rather than facts. Furthermore, the consistency of terms can be improved. The most prominent example being the term "model", which is sometimes apparently confused with "model component", "model configuration", or "experiment" (see below in specific comments).

Overall, the manuscript is considered well written, despite this criticism.

**Specific comments**

p2l2: The claim that ESMs are written for specific purposes and lack modularity seems not well founded and seems to contradict the fact that ESMs are often composed of (independent) component models. I do agree with the implication that users need profound knowledge and time to learn a new ESM, but, in my opinion, this is due to complex configurations and lack of configuration management options. There is, often, a lack of modularity *within* component models, but this is not addressed by ESM-Tools either.

p2l39: It is not clear what "script-based coupling" is.

p2l44: The list of tools seems very selective. Without having a good overview of ESM infrastructure tools, it strikes me that three of the four listed tools are from Germany.

p3l68ff: The description of Mkexp seems rather subjective. The language (e.g. "lengthy", "unintuitive") indicates opinions rather than facts.

p3l83: It is not clear what the "specified criteria" are or where they are specified.

p4l94: The sentence "To achieve this, the code ... is organised such that all information about the model (e.g. HPC systems, ...) ..." does not make sense to me. First, it seems

to me that not the "code" is organised, but that it is the concept of ESM-Tools to provide information in YAML files. Second, the "HPC system" is not, in my view, information about the "model". I would argue that the HPC system is a property of an experiment, or even more specific, one experiment run. You could have the same model in various experiments on different HPC systems.

p4l96: It is not clear what the term "re-coded" is referring to. Is it previous versions of ESM-Tools?

p5l113: Again, it is not entirely clear to me what the term "model setup" refers to. Is it the model configuration (e.g. components, grids, ...)? Or does it include experiment parameters?

p5l124ff: Maybe this part of the section should not be a bullet point list.

p6l131ff: The detailed workflow for pulling component models from their respective repository and putting them into a coupled model is not clear. It says that "compile time switches" are activated but that means that the components must be prepared for the coupling in the first place. This would be reasonable assumption, but what if further changes are needed? In some way, the section explains the downstream view of things ("huge improvement for model developers") but not upstream or deliberate isolated contributions.

p6l43: Does the sentence indicate that dependencies are "baked into" ESM-Tools? That would indeed make installation easier, but it has obviously disadvantages as well (e.g. outdated dependencies). If external packages are included with ESM-Tools (and the fact mentioned in the article), it should be mentioned which and why.

p7l165: It is not clear which metric "most extensive" refers to. Also, "optimized implementations of all the functionalities required to run an ESM" confuses me. Which functions are meant and in which ways are they optimised?

p8f2: What are "user defined functions", more specifically, who is the "user" in this

case? Is it the user that runs an experiment or the developer that prepares one model (configuration) for use with ESM-Tools?

p8l174: It would be good to explain how esm_viz interfaces to the raw model output for monitoring. An example YAML file/snippet illustrating the monitoring configuration would be helpful.

p8l188: What is the "modular layout of the repositories used by the ESM-Tools"? Does it refer to the ESM-Tools code or the repositories of the models (model components)? How does the GUI make use of the modularity?

p9l191ff: It would be nice to give an overview, either in this section or earlier, about the specific steps needed to port a new model to ESM-Tools.

p11f4: It is not clear what information is given (above Table 2+3) and if the colours used in the figure have a specific meaning.

p12l231ff: Since the contribution workflow seems to differ from the standard Github clone-pullrequest scheme, it could be more detailed how users "develop their own extensions to the existing code". If(!) I understand correctly, the actual code is not hosted on Github, so a clarification could be needed.

p12l298: It should be made clearer what part of ESM-Tools is hosted at Github (and covered by the GPLv2 license?) and what "development access" means. It seems that the software at Github comprises mostly the configurations for models/components and platforms, but not the actual code.

**Technical corrections**

p7f1: The upper part of the figure is about the build process, but I'm not sure if the "esm_master get-model" command is consistent with that. I would expect something like "esm_master comp-model" or similar. But that's without knowing the ESM-Tool commands.

---

## Author Comment (AC1) · 24 Dec 2020

Dear Sir/Madam,

First of all we would like to thank you for the detailed comments you provided for our manuscript, they were much appreciated and we hope that we were able to improve the paper by carefully re-writing the mentioned sections. We especially tried to give a clearer overview, partly by adding extra explanations to the description chapter, partly by re-arranging paragraphs from the discussion to earlier parts. Also, we included several more examples of YAML configuration files, to make it clearer what we are talking about.

[Figure]

Concerning the detailed comments, let me shortly answer those point by point:

\*\*SPECIFIC COMMENTS\*\* (P: page, L: line or lines) Agreed, we added requested reference. Agreed, we tried to make that clearer by explicitly mentioning that ESM-Tools does not help with the coupling at all, but with the application of already coupled models. Agreed, we elaborated on provenance tracking and reproducibility. 4. We rephrased that to make it clearer. Additional comments to different parts of the manuscript:

"Abstract" - P1 L4f: The reason is that even though starting from the same model component, scientists develop their own version with additional features, that might or might not end up back in the development trunk. E.g. echam-wiso, echam with concurrent radiation, etc. Added a sentence to the abstract. - P1 L5f: Changed sentence "This procedure not only leads to a strongly growing number of available versions of model components and coupled setups, but also to model- and system-dependent ways of obtaining and operating them." to "This procedure not only leads to a strongly growing number of available versions of model components and coupled setups, but also to model- and HPC-system dependent ways of obtaining, configuring, building, and operating them." - Agreed, done. "Introduction" - P1 L20: Changed sentence "Earth System Models (ESMs) are widely used for studying past, present and future climates. They can include several model . . ." to "Earth System Models (ESMs) are widely used for studying past, present and future climates, processes in the different Earth system compartments (e.g., atmosphere and ocean), as well as the interactions between them. Therefore, ESMs can include several model ..." - P1 L21: Changed list "atmosphere, ocean, land and sea ice, land biosphere, ocean biogeochemistry" to "atmosphere, ocean, ocean biogeochemistry, land and sea ice, land biosphere, hydrology" - P2 L23: Moved the sentence "Since the ESMs are usually written for a specific purpose, they often lack modularity." a bit further down. And changed it to: "Since the ESMs are usually written for a specific purpose that e.g. require a specific coupling and/or tailored ESM components, they often lack modularity.. This lack of modularity ..." - P2 L48: Added the mentioned reference and cited in line 49. - P2 L44ff: Added an list item for ESMF: "\item The \textit{Earth System Modeling Framework}, ESMF\footnote{\url{http://earthsystemmodeling.org/}} is a standard software platform for Earth system models \citep{Hill2004}. It provides different structures for the interconnecting between model components and provides a standard support library for the construction of components. The emphasis of ESMF is to ensure a standard infrastructure of component coupling and may require code adaptation in order to fit into its framework." - P3 L83: I changed the sentence "The ESM-Tools emphasize the needs of researchers for such a software and fills this gap by fulfilling the specified criteria." To "The ESM-Tools emphasize the needs of researchers for such a software and fills this gap by fulfilling the criteria such as user-friendliness, modularity, portability, maintainability, extendablilty (see also Section~\ref{sec:yamlconf})."

"ESM-Tools Description" - Moved the part from the discussion here, and added a more detailed overview. - P4 L93ff: See comment above, tried to make that clearer in the text and with examples. - P4 L98: Rephrased that. - P4 L110: Explained that in more detail, and added a YAML snippet - P5 L113ff: Elaborated on that, and included another YAML example - P6 L137ff: Explained that in detail, also added that to the "modular" section. Added the sentence: "In contrast to other software described above, e.g. MESSy or ESMF, ESM-Tools are designed to help scientist to build and run different standalone models as well as coupled setups without any need of code adaptation. To ensure this, all ESM components need to have a dedicated coupling already implemented." on page 4 line 88. - P7 Fig1: We explained that in the first part of description now. - P8 Fig2: Nah, I like the figure, but we added an explanation to the text of it - P7 L165ff: I don't think that esm_runscripts and the mentioned workflow tools are comparable. What esm_runscripts tries to do is to provide a common interface - common "runscripts" so to say, even if no scripting is involved anymore - to a growing number of model components and coupled setups, to make the application of ESMs easy and unified across the ESM community. Even though we naturally try to cover more and more of the workflow involved, we are far from workflow managers like the ones mentioned by the referee, of which we will be able to learn much in the future. - P8 L175f: We referred to a future publication, that will include more topics on data processing with ESM-Tools. - P8 L185ff: The tool esm_master does, and the GUI just displays the information esm_master has anyway. - P9 L192ff: see above, comment P6 L137 - P9 L200: Elaborated on that a bit, not sure if I understood the comment correctly. - P10 Tab3: We added a column for the coupler of the setup. - P11 Fig4: We deleted the Figure 4 (old numbering) and removed the sentence: "Figure~\ref{fig:models} illustrates how these models can be coupled together, resulting in various models and coupled systems which are supported by the ESM-Tools." Page 9, line 198. - Done that, added several more examples. "ESM-Tools Development" - P11 L209: No, we don't. Even following SCRUM is a bit over-the-top for this, we call our approach SCRUM-light... Maybe a development plan should be implemented iin the future. Added the sentence "The ESM-Tools software is currently hosted and developed on GitHub (see also \textit{Code availability})." Also: Can you provide any recommendations in software development, which are worth being shared? Not really, except what is mentioned in the paper already. In out opinion, a structured project management is fundamental to a successful software development. In scientific context though, the need to react to user problems, changing HPC environments, new model developments etc makes it really difficult to go "full SCRUM", which is why we adapted the philosophy of SCRUM, while not taking it too seriously at times. SCRUM light, if you want to call it that way. "Discussion" - P13 L252ff: Done. - P14 L295ff: Agreed and done.

I hope we could include all suggestions of the referee, and want to thank him/her again for the kind help. Dirk Barbi, for the ESM-Tools team

---

## Author Comment (AC2) · 24 Dec 2020

Dear Sir/Madam,

First of all we would like to thank you for the detailed comments you provided for our manuscript, they were much appreciated and we hope that we were able to improve the paper by carefully re-writing the mentioned sections. We especially tried to give a clearer overview, partly by adding extra explanations to the description chapter, partly by re-arranging paragraphs from the discussion to earlier parts. Also, we included several more examples of YAML configuration files, to make it clearer what we are talking about.

[Figure]

Concerning the detailed comments, let me shortly answer those point by point:

**Specific comments** p2l2: Rephrased that to make it clearer.. p2l39: Explained that, and added Paul Gierz's SCOPE paper as reference p2l44: The author is right, and I tried to explain why. p3l68ff: Explained that this is our experience as model supporters, avoided the criticized language p3l83:Rephrased that: Changed the sentence "The ESM-Tools emphasize the needs of researchers for such a software and fills this gap by fulfilling the specified criteria." To "The ESM-Tools emphasize the needs of researchers for such a software and fills this gap by fulfilling the criteria such as user-friendliness, modularity, portability, maintainability, extendablilty (see also Section~\ref{sec:yamlconf})." p4l94: Addressed the comment by adapting the paragraph "To achieve this, the code of the ESM-Tools is organised so that all information about models (e.g., HPC systems, input datasets) are contained in YAML\footnote{\url{https://yaml.org/}} configuration files, while the actual program, i.e. the commands to be performed using this information, is in itself entirely independent of the models or HPC systems used, and has been fully re-coded in Python." Changed to "To achieve this, the software architecture of the ESM-Tools is structured so that all information that are mandatory for an experiment (e.g., HPC systems, input datasets) are contained in separate YAML\footnote{\url{https://yaml.org/}} configuration files, while the actual program, i.e. the commands to be performed using this information, is in itself entirely independent of the models or HPC systems used. For this purpose, the ESM-Tools software has been fully re-coded in Python (see also Section~\ref{sec:python}) compared to previous versions." p4l96: Removed the reference to a previous version due to other referee's comment p5l113: Made that clearer, meant coupled setup. p5l124ff: Changed LaTeX 'itemize' environment to 'description' environment. p6l131ff: Added a figure and explanation p6l43: If external packages are included with ESM-Tools (and the fact mentioned in the article), it should be mentioned which and why. That is just how python works. . . Yes, we include expernal packages, but a well-written setup.py makes sure that all dependencies are met during the installation process. I don't agree with the statement that a list of packages should be

included as it doesn't add information about our product - I don't think the reader needs to know in this instant which YAML reader we use, for example. And there is always the documentation for these detailed questions.... p7l165: Added the half sentence "...in terms of functionality." Changed the sentence: "This tool consists of optimized implementations of all the functionalities to run an ESM." to "This tool consists of optimized implementations of all the functionalities to set up all required phases to run an ESM simulation."

p8f2: added an explanation on the caption of the figure p8l174: Included reference to a future publication p8l188: This refers to the repositories of the supported models. Added "... of the models" here. p9l191ff: Added some text, also two figures with yaml p11f4: It is supposed to show that we warp different compilation strategies.

p12l231ff: This is a misunderstanding. The code is hosted and developed on Github. Added the sentence on page 11, line 214: "The ESM-Tools software is currently hosted and developed on GitHub (see also Code availability).". Also changed paragraph at page 12, line 135 by adapting the sentence to "This can be done by contributing to the development on GitHub (e.g., by pull requests). Furthermore, users can suggest a "wish-list" of missing functionalities (e.g., via issue tracker)." p12l298: This is a misunderstanding. Both, code and configuration are hosted and developed on Github. See also comment above. Made that clearer in the text. **Technical corrections** p7f1: Changed figure: esm_master get-model1 -> esm_master install-model1 and added other tasks.

I hope we could include all suggestions of the referee, and want to thank him/her again for the kind help. Dirk Barbi, for the ESM-Tools team

---

## Author Response (AR2)

Dear referees,

Thank you for your kind and detailed remarks on our manuscript. We tried to resolve all open points and correct the text. I would like to answer in detail to your comments:

To referee 1:
> Title:
Since version 5.0 of ESM-Tools has been released, I wonder if it would make sense for the authors to document the latest version, provided that the upgrade from 4.0 to 5.0 is not too dramatic, such that a significant part of the text needs to be updated/revised. This is up to the authors to decide. -> Done

> I'd suggest the authors state upfront that ESM-Tools is not a coupling software early in the Introduction and/or Abstract. I thought (or at least was in doubt) it has this capacity until I read later in the manuscript. Such a statement can be added at, for example P1L18 and P2L37. -> Done

> P1L11: …the ESM-Tools "are"…
The authors treat the ESM-Tools randomly either as singular or plural throughout the whole manuscript. Please make it consistent (I think it should be a singular noun). -> Done

> throughout the text, the authors use inconsistent references to the figures, e.g. fig x, fig (x), Figure x. Please make it consistent and comply with GMD format if any. -> Changed that to be GMD compliant, which is Fig. 3 in  the middle of a sentence, Figure 3 at the beginning.

> P1L12-13: In fact, … standalone simulations.
I don't see this sentence as important info in the abstract, as the previous sentence has mentioned this point. If the authors decide to have it, more information on FESM2, ECHAM, ICON is needed, for example at least on what components they are (ocean or atmosphere or). Full names are usually needed but that would make the abstract tedious. -> Done

> P2L29-30:
I don't quite understand when the authors say that 'ESMs are usually written for a specific purpose'. It reads ambiguously to me; do the authors mean 'configured for a specific purpose'? -> tried to write that more clearly

> P2L45:
I don't understand what 'script-based coupling' is. -> Changed to "and off-line coupling through separate scripts"

> P3L71:
Do the authors mean 'nesting' of model components here? Since it is a tool for coastal ocean modelling, it should only have the ocean model component, right? -> Yes.

> Figure 1:
I don't see the point to have this as a figure. These are very basic lines and the description in the text is sufficient. In addition, the caption is a bit colloquial, e.g. it could be 'Some basic

requirements need to be met before…' -> Changed the sentence, would like to keep the figure as it demonstrates how easy it is to install, which, annoyingly, is one of the mayor objectives when we try to convince scientists to try it.

> P6L148:
I agree the tool would save lots of time for researchers, but I don't see how the tool would have the 'potential of reducing model diversity'. -> removed that

> P6L154:
This sentence seems to be coming out of the blue, especially 'solving a model run error'? -> Removed it.

> P9L204-244:
This part of the text does not fit the scope of this subsection (2.2.2 YAML Configurations) and should be moved elsewhere, for example in the discussion. -> I don't agree, as this is the main motivation for using yaml rather than shell scripting. Would like to keep it here.

> I also find the tool very useful for educational purposes and for provenance/reproducibility of model simulations. The authors have mentioned the latter, but perhaps that can be stressed even more. -> Definitely, but as that wasn't anything we have experience with already, we skip that to a later time.

> P9L215:
I don't understand this 'compile time switches'. -> Changed to "preprocessing options"

> Table 3 seems to be not referred to in the text at all. -> Done

> P12L270-271:
This is up to the authors to decide if they think it could be useful: it would be good to elaborate a bit more on the tool's log file and post-processing capacity. For example, for the log files, does the tool collect all the log info from each component and put/condense them together, or most log info still stays with their respective component? For the post-processing, does the tool offer things like netcdf conversion, climatology/time series calculation, or even cmorisation to name a few? The feature of online automatic cmorisation would be very useful for many. -> As this is partly discussed right at the moment, we would not like to elaborate too much on that point right now. We applied for a project to work on exactly that, but alas, that will not be funded - so we are unsure about the strategy, or even funding.

P12L275-276:
I like this esm_vis feature a lot, but I don't agree on this point, since modellers can always sanity check the simulations anytime without having to wait for the whole run to be completed. I think the main advantage is that it is very convenient/handy to use and save modellers' time. -> Removed the sentence part ", since particularly for long-running jobs waiting for the job to complete (which may take several weeks) before noticing any errors can be avoided."

> P19L387-398: these two paragraphs are almost identical to the first two paragraphs in this section? Please check. -> Oops. that happened when restructuring the text. Removed the second paragraph.

Technical corrections:

> P1L20: move 'lately' up in the sentence, e.g. 'Compared to previous versions, ESM-Tools has lately been …' -> Done

> P2L42: The abbreviation HPC should come earlier. It does already, L 11.

> P2L51: please give the full name of SMHI. -> Done

> P2L55: 'This approach is also…' -> Done

> P3L67-68: should be updated; also see my first comment. -> Done

> P3L78: 'in our opinion it lacks…' -> Done

> P3L84: remove 'the' -> Done

> P4L89: please fix the ref for Righi et al. -> Done

> P4L101: 'The tool's application…' -> Done

> P4L105: '…applying an ESM…'; the full name has been defined at the beginning of the manuscript. -> Done

> P5L119, 122: 'dependent' -> Done

> P5L127: remove 'are' -> Done

> P5L137, and elsewhere in the manuscript: no space before and after '/' -> Done

> P5L138: '…, and some features will be…' -> Done

> P7L161: should the footnote to YAML come earlier in the draft where it first appeared? -> Done

> P7L163: '…, while the ESM-Tools…' grammar issue; please consider starting a new sentence. -> Done

> P7L174: 'organize'? -> Done

> Table 1, in the row of esm_calendar: ''but also works…' -> Done

> P8L178: 'to alter' -> Done

> P8L180: 'this dictionary' -> Done

> P8L181: 'execution of a compute job' -> Done

> Figure 3 caption: 'Section of…' -> Done

> P8L192: 'to generate' -> Done

> P8L192: 'and in this sense can enable…' -> Done

> P10L233: please try to avoid words like 'nicely' -> Done

> P11L242: 'for us', the authors mean 'core developers'? and the previous 'developers' are 'contributing developers'?-> Done

> P12L279: please avoid using 'you', 'your'. This also applies to P16L339 and P17L346. -> Done

> P12L280: 'the remainder of the supercomputer'? not clear to me -> to me neither, removed that

> P14L307: 'and the code' -> Done

> P16L316: 'the tool, which…' -> Done

> P16L319: remove 'even'? -> Done

> P16L342: 'configuration'; 'As an example, …' -> Done

> P17L347: 'see an example again from…' -> Done

> P18L355: '… Project phase 6, CMIP6, …' -> Done

> P18L371: ', which were discussed…' -> Done

> P18L372: 'point out that…' -> Done

> P18L378: 'as if it were…' -> Done

> P19L383: ', as we provide…' -> Done

> P19L407: 'webpage' -> Done

> P20L409: 'upcoming' -> Done

To referee 2:

I am happy with the changes that the authors made; I think that the revised version of the manuscript is much improved. The level of detail provided has been risen, the structure is better now, which aids in the understanding, also due to additional text such as in Section 2.1. My remaining comments are minor and listed below. I also recommend to check grammar, punctuation, and consistency in detail. Apart from the remaining minor issues, I would recommend a publication of the manuscript.
With kind regards.

L16: Perhaps add something like "once implemented with ESM-Tools" -> rephrased the part due to other referees comments

L34: "requires" -> Done

L98: Remove the bullet point, as this is not part of the listing. -> Done

L103f: Make this statement even stronger; it has to be emphasised ESM-Tools aid not in the coupling as such; just to avoid any misunderstanding -> Added a remark to the abstract to make it obvious

L131: Add the type of model and perhaps a reference. -> Added references

L136: right -> access privelidges -> Done

L137: Very specific coupled setups are supported, right? -> Yes

L138: What kind of conflict is meant? -> added " between standalone and coupled setup settings"

L190ff: Thanks for including this paragraph, but perhaps revise a little to make it briefly more clear, how the user defined functions are embedded in with ESM-Tools. -> in principle we showed that by the recipe, anything more would quickly go into the very technical bits.

L219ff: The listing of features is important and reads nicely, albeit I am wondering whether this is not a separate aspect, rather than a part of the Section 2.2.2 on the YAML configuration files; it reads rather like a section on "efficient maintenance" -> I don't agree, as this is the main motivation for using yaml rather than shell scripting. Would like to keep it here.

L382: Perhaps mention some publications whose model runs were supported by ESM-Tools -> Done

L440ff: There is some redundancy and doubling of text it seems in the last two paragraphs of the discussion. -> Removed the second paragraph

---

## Author Response (AR3)

Letter to the Topical Editor

Dear Ignacio Pisso,

Thank you so much for your quick response, and the details on how to get to a finalized manuscript. We sincerely hope that it can be published soon now, before our software goes into release 6 and we can start with adapting the paper all over again.

In detail, here is our response to the corrections you listed:

L59
both software package can be seen as natural
-> both software packages can be seen as natural  → done

L66
"and have implemented first steps towards integrating full MESSy support into ESM-Tools, hoping to complete this process by release 5.0 (planned for October 2020)." This paper is supposed to already describe v5.0 now.  → removed the date, as not much interest is shown in MESSy support at the moment.

L108:
The ESM-Tools software is devided into three major parts
-> The ESM-Tools software is divided into three major parts
This was pointed out by the reviewer but not modified. → Done, sorry about that.

L172:
We separated our (Python)
I think the issue is to use present and not past tense → Still don't see what's the point, but changed to present.

2.2.1 has lost a whole paragraph while the corrections by rev 1 are marked as done. Was intentional? → Yes, as the reviews were quite extensive, we reworked that part to make it more concise, and better focused on out tools, rather than Python features.

From reviewer #1
> P16L342 (in version 3 of the manuscript): 'configuration'; is Done but 'As an example, ...' is not Done.
(in version 4: ...'the model. An an example, in Fig. 9,...') → Corrected that

Please correct the alignment of appendix subsections with the corresponding figures.
A1 Example ESM-Tools runscript
A2 Example YAML configuration file → changed the width of one figure and introduced a couple of newpage commands to make them fit nicely

I hope that now, finally, we have met all requirements for publication,

Best regards,
Dirk Barbi for the ESM-Tools team